# Atypical flagella assembly and haploid genome coiling during male gamete formation in *Plasmodium*

Molly Hair [1], Flávia Moreira-Leite[1], David J. P. Ferguson[1], Mohammad Zeeshan [2], Rita Tewari [2] ✉ & Sue Vaughan [1] ✉

Gametogenesis in *Plasmodium* spp. occurs within the *Anopheles* mosquito and is essential for sexual reproduction / differentiation and onwards transmission to mammalian hosts. To better understand the 3D organisation of male gametogenesis, we used serial block face scanning electron microscopy (SBF-SEM) and serial-section cellular electron tomography (ssET) of *P. berghei* microgametocytes to examine key structures during male gamete formation. Our data reveals an elaborate organisation of axonemes coiling around the nucleus in opposite directions forming a central axonemal band in micro-gametocytes. Furthermore, we discover the nucleus of microgametes to be tightly coiled around the axoneme in a complex structure whose formation starts before microgamete emergence during exflagellation. Our discoveries of the detailed 3D organisation of the flagellated microgamete and the haploid genome highlight some of the atypical mechanisms of axoneme assembly and haploid genome organisation during male gamete formation in the malaria parasite.

The prerequisite for the transmission of *Plasmodium* spp. from an infected mammalian host to *Anopheles* mosquito vector relies upon the parasite switching from asexual to sexual reproduction. This maturation occurs when an asexual schizont differentiates into a sexually committed merozoite which gives rise to immature male and female gametocytes within the blood of the intermediate host[1–3]. Only on ingestion of blood meal by an *Anopheles* mosquito do these gametocytes undergo development to produce gametes, both male (microgamete) and female (macrogamete)[3–5]. Within the mosquito's midgut, environmental changes of pH, the presence of xanthurenic acid and a drop in temperature are stimuli that activate the micro-and macrogametocytes. Post activation, microgametocytes undergo three rounds of atypical intra-nuclear mitosis from haploid (1N) to octoploid (8N) and assemble eight axonemes within the cytoplasm of the microgametocyte, without intraflagellar transport. The microgametocyte then undergoes exflagellation to release eight haploid motile microgametes capable of fertilising macrogametes within 15 min[5–7].

Coordinating this event are bipartite microtubule organising centres (MTOCs), each composed of an acentriolar MTOC (called a nuclear pole NP), which lies on the inner nucleoplasmic side of the nuclear envelope and is responsible for spindle microtubule formation, and a centriolar MTOC (called a basal body BB) associated with an axoneme on the outer side of the nuclear envelope. The cellular ultrastructure of nuclear poles (NPs) and basal bodies (BBs) have previously been observed by electron microscopy as two separate electron-dense structures, bridging between the nuclear and cytoplasmic sides of the nuclear envelope[5,7,8]. During the early stages of microgametocyte activation (1–3 min), the inner and outer MTOCs work initially to coordinate mitotic division and to nucleate eight axonemes with rapid axoneme assembly occurring within the cytoplasmic region around the single nucleus[2,6,8,9]. By 6–8 min post-activation most axonemes have formed and by 8–10 min exflagellation has begun with basal bodies protruding first out of the cell membrane, each one bringing with it an assembled axoneme and a

[1]Department of Biological and Medical Sciences, Oxford Brookes University, Gipsy Lane, Oxford OX3 0BP, UK. [2]School of Life Sciences, University of Nottingham, Nottingham NG7 2UH, UK. ✉e-mail: rita.tewari@nottingham.ac.uk; svaughan@brookes.ac.uk

haploid genomic nucleus, the essential components of motile microgametes[8,10]. The whole process is usually complete by 12–15 min[2,5,9].

Visualising the process of microgamete assembly and exflagellation by thin-section transmission electron microscopy (TEM) only provides 2D views of the ultrastructure, but it is clear from live cell studies that microgamete assembly is a complex three-dimensional process[5,9]. Here, we have used serial block face scanning electron microscopy (SBF-SEM) to reconstruct whole individual microgametocytes and key elements in individual microgametes. In addition, we performed serial electron microscopy tomography of individual microgametes, which revealed interesting and distinctive features of their ultrastructure. The combined volume electron microscopy data presented here improves significantly our structural understanding of the dynamic process of *Plasmodium berghei* male axoneme assembly, flagellar formation and haploid genome organisation within the microgamete.

## Results and discussion
### Analysis of basal bodies and nuclear poles reveals heterogeneity in association during male gametogenesis
*Plasmodium berghei* purified gametocyte cells isolated from infected mouse blood were activated for 15 min and then fixed, embedded, and imaged by SBF-SEM. This process resulted in two datasets, both containing 197 individual slices (Z-slice = 100 nm) (Supplementary Movie 1). In order to specifically visualise the three-dimensional organisation of axonemes, microgametocytes containing assembling axonemes were chosen for further analysis. From these datasets 15 individual whole microgametocytes with their axonemes internally coiled were identified and analysed. This represented the total number of complete microgametocytes and microgametes in our datasets. The plasma membrane, nuclear envelope, nuclear pole MTOCs (NPs), basal body MTOCs (BBs) and axonemes were segmented for each whole cell using IMOD[11] to produce 3D models (Supplementary Fig. 1).

*Plasmodium* basal bodies do not exhibit a defined 9 triplet microtubule ultrastructure by TEM as seen in many other organisms. Their BB structure is defined as an electron-dense mass where, on rare occasions, nine single A-tubules can be resolved on the cytoplasmic side of the nuclear envelope, with the NP (from which spindle microtubules radiate) closely associated on the inner surface of the nuclear envelope[8,10,12]. By following along the length of each axoneme in our 3D reconstructions, it was possible to locate the BB and NP closely associated with the envelope of the nucleus. The BB and NP are physically cross-linked through the nuclear envelope[8,12,13] and in our SBF-SEM data slices electron-dense filamentous material can be observed in most cells between the two MTOCs (Fig. 1A; arrowheads).

Of the 15 microgametocytes analysed, 12 contained 8 BBs with axonemes and 8 NPs. The resolution in the remaining 3 microgametocytes was limited; therefore, we were unable to clearly distinguish the NPs, but they did contain 8 BBs with axonemes. Moving through the SBF-SEM data slices of the 12 microgametes where all NPs were visible allowed us to identify which BB was in close proximity to which NP. This analysis revealed that the majority (75%) of BBs and NPs were physically adjacent (Fig. 1B). Nevertheless, there were BBs with assembled axonemes that were not associated with a NP and there were NPs that were not associated with a BB (Fig. 1B, C). For example microgametocyte cell 1 shown in Fig. 1B clearly shows 2 BBs without an NP in close proximity and 2 NPs without a BB in close proximity (Fig. 1C). We measured the distance between each associated BB and NP in each of the 12 microgametocytes using the mid-point of each electron dense structure to quantify their relationship. Each NP and BB typically spanned 2–3 (BB) or 3–4 (NP) z slices. For each structure, we estimated the mid-point as being the centre of the electron density of each structure. Our data revealed the average distance between the two was 431 ± 128 nm (Fig. 1D; n = 71 basal body/nuclear pole pairs). It is important for BBs and NPs to remain connected so that an axoneme can be associated with a single haploid microgamete nucleus upon exflagellation, but they must become disassociated at some stage so that exflagellation can occur. These disassociated BBs and NPs observed at this stage represent aberrant forms and are unlikely to support genome segregation into a budding microgamete. This is in agreement with data showing that male gametogenesis does not always develop correctly[5,13,14].

Next, we analysed the positioning of the 8 NPs within the nuclear envelope. As the cell undergoes three rapid rounds of DNA replication, spindle microtubules nucleate from each NP and chromosome segregate as observed with kinetochore marker NDC80[15–17]. The basal body marker SAS4-GFP is found associated with the kinetochore marker NDC80-mCherry even at 8 min after activation (when the nucleus is octoploid), as observed by live cell imaging (Fig. 1E)[8]. In early stages, NPs are often located in closely associated pairs. As gametogenesis progresses NP pairs are located around the circumference of the nucleus before each pair finally separates and 8 single NPs are separated around the circumference of the nuclear envelope[5,10]. By SBF-SEM we found heterogeneity in the positioning of the NPs among microgametocytes at pre-exflagellation stages, which strongly suggests that NP movement is occurring whilst axonemes are forming (Fig. 1F–H). In 4 cells the nuclear pole pairs were grouped on one side of the nucleus (Fig. 1F). In 5 cells we saw the pairs of NPs positioned in opposite sides of the nucleus (Fig. 1G) and in 3 cells, most of the NP pairs were grouped together, with a single pair on the opposite side of the nucleus (Fig. 1H). Taken together, these results show variations in the 3D organisation of BBs and NPs, and demonstrate the close association between BB and NP during male gametogenesis.

### Axonemes coil around the nucleus in opposing directions creating an axonemal band
A hallmark stage post activation of microgametocytes is the assembly of axonemes which are internally coiled around the nucleus (Fig. 2A), and can clearly be seen at ~3–4 min after activation using live cell imaging microscopy and basal body marker SAS-GFP with an axoneme marker Kinesin8B-mCherry[18]. Our SBF-SEM datasets have allowed further detailed analysis of this coiling, which we term an axonemal band. In the 15 microgametocytes analysed each had 8 axonemes internally coiled around the nucleus and in close association with one another (Fig. 2B). Using the BB as an orientation marker, we discovered that axonemes do not all coil around the nucleus in the same direction. Each axoneme was segmented in yellow or blue depending on their orientation around the nucleus (Fig. 2B, Supplementary Movie 2). We found that the two axonemes of each BB/NP pair coil in the same direction, suggesting a level of positional constraint (Fig. 2C, D). We found only one occasion where the two axonemes of a BB/NP pair were coiled in opposite directions. Taken together, these results suggest there is a level of asymmetry/chirality in the ultrastructural organisation of BB/NP/axoneme pairs in the microgametocyte. This asymmetry may be determined by the BB structure or mitotic spindle organisation, ensuring that each pair coil around the nucleus rotates in a defined direction. Presumably, the formation of this axonemal band could be advantageous in preventing intertwining of beating axonemes inside the cell and during exflagellation.

### Outer doublet microtubule assembly precedes central pair formation in elongating axonemes
The unique SBF-SEM 3D datasets we produced had sufficient resolution to allow us to analyse and quantify axoneme assembly in each microgametocyte. The length of each axoneme in a cell was measured and was found to be highly variable in all cells ranging from 3 μm to 20 μm, with an average of 13.6 ± 0.9 μm (Fig. 2E). There was sufficient resolution in these datasets to distinguish clearly the outer doublets of a 9 + 2 axoneme as an electron dense circle, with the central pair

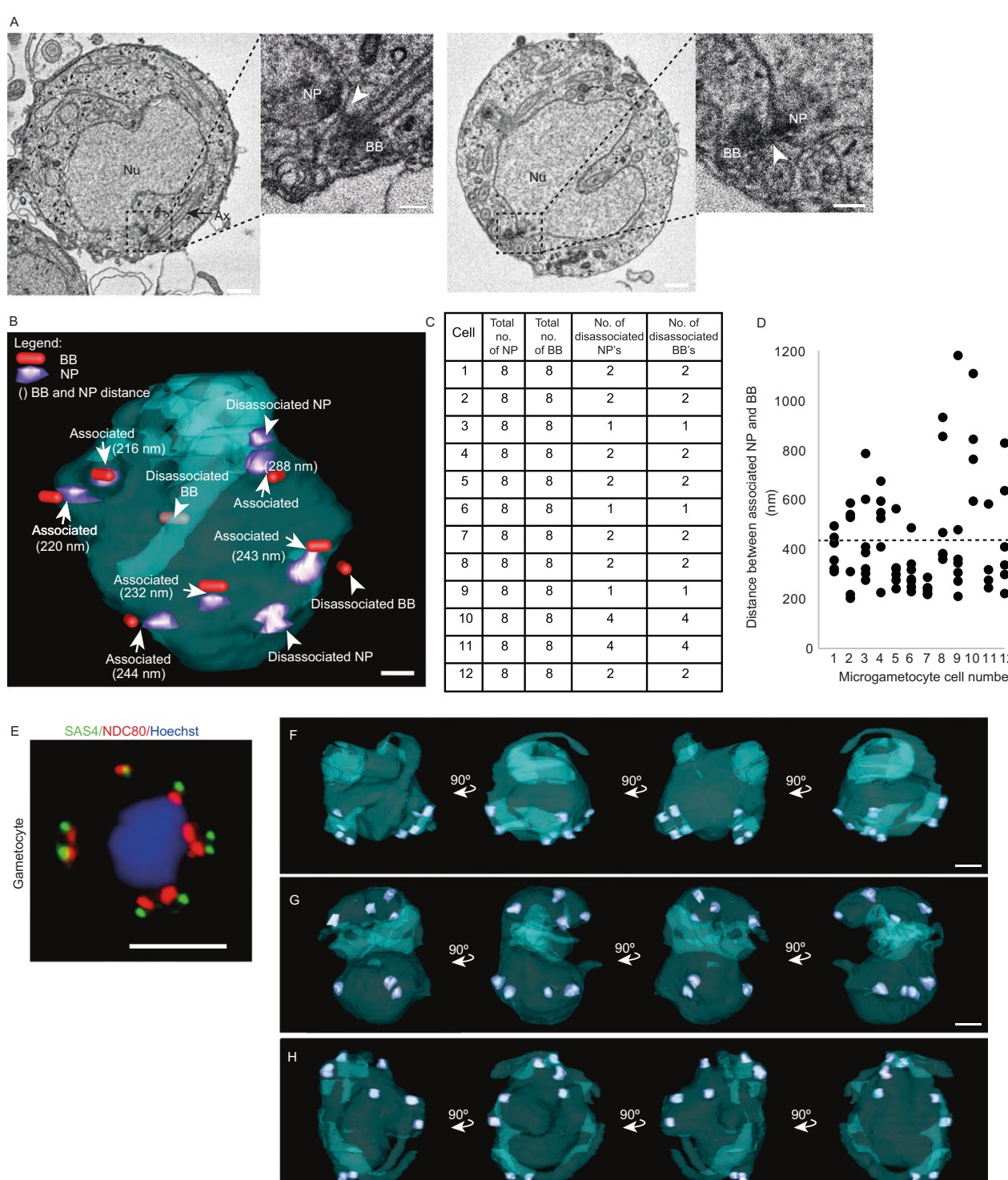

Nature Communications | (2023)14:8263                                                                                   **3**

appearing as a central electron density within this circle (Fig. 2F; $z = 0$ nm). This then allowed us to analyse outer doublet and central pair assembly, as sections without a central pair are easily visible (Fig. 2F; $z = 100$ nm) and could reveal the progress of axoneme assembly at this pre-exflagellation stage. Assembly of outer doublet microtubules has been shown to precede central pair microtubules during axoneme growth in the Kinetoplastida parasite *Trypanosome brucei* by ~50 nm, but in fully assembled older axonemes central pair microtubules extend to the distal end[19]. All 8 axonemes had begun assembly in each of the 15 microgametocytes (Fig. 2E; $n = 120$ axonemes). However, all microgametocytes contained axonemes that did

not have fully elongated central pair microtubules and only 40 out of 120 axonemes had a fully elongated central pair (Fig. 2E). Using these criteria we conclude that most axonemes are still in the process of assembly at this stage in male gametogenesis. This finding is an important aspect to take into consideration when assessing the timing of gametogenesis and analysing axoneme mutants.

### Unique organisation of the nucleus around the axoneme of microgametes

The nucleus was reconstructed in each cell and this revealed a highly contorted 3D structure (Fig. 3A; two cell examples; $n = 15$ cells). There

**Fig. 1 | Analysis of basal bodies and nuclear poles reveals heterogeneity in their association.** A total of 15 microgametocyte cells 15 min post-activation were analysed and 3D reconstructed from SBF-SEM data of gametocyte populations, to examine the association between basal bodies (BBs) and nuclear poles (NPs). **A** SBF-SEM data slices from two microgametocytes showing two examples of the typical bipartite MTOC composed of a basal body (BB) and a nuclear pole (NP) linked by filamentous material (white arrowheads). Nu, nucleus; Ax, axoneme (black arrow). Scale bar = 1 μm in lower magnification images, or 200 nm in insets. $n = 15$ microgametocyte cells. **B** A 3D segmented model of a microgametocyte nucleus (blue) with associated and disassociated BBs (red) and NPs (purple) indicated. The distances between associated BBs and NPs in each pair are also indicated (nm). Scale bar = 500 nm. **C** Table showing the total number of NPs and BBs per microgametocyte cell, and the total number NPs and BBs per cell that are disassociated (i.e., not linked forming a bipartite MTOC). **D** Dot plot showing the distances (nm) between the BB and the NP in associated pairs, in each microgametocyte cell. Average distance (431 nm) is indicated with a black dashed line. Source data are provided as a Source Data file. **E** Fluorescence microscopy image showing the localisation of the BB marker SAS4-GFP (green) in relation to the NP (kinetochore) marker NDC80-mCherry (red) in male gametes 8 min post-activation. Scale bar = 5 μm. This experiment was repeated three times with similar results. **F–H** Segmented nuclei (blue) at 90-degree rotations showing the three different configurations of nuclear pole (purple) positions observed across the 15 cells. **F** Nucleus showing all nuclear pole pairs positioned at one side of the nucleus, (**G**) Nucleus showing nuclear pole pairs positioned in opposite sides of the nucleus and (**H**) Nucleus showing nuclear pole pairs grouped together, with a single pair on the opposite side of the nucleus. Scale bar = 5 μm.

were multiple protrusions with BB/NPs often positioned at the end of these protrusions (Fig. 3B; arrow). Measurements of nucleus volume revealed some variation in volume with an average volume of $27 \pm 3.4 \, \mu m^3$. Our datasets also included 5 microgametocytes further along the process of gametogenesis, where individual axonemes were found to be in the process of exflagellation (Fig. 3C–G, Supplementary Movie 3). Not all microgametocytes at this exflagellation stage contained 8 internal axonemes; thus, it is likely that some axonemes had already ex-flagellated in these cells. For example, in one microgametocyte there were 2 microgametes undergoing exflagellation and no internal axonemes, suggesting that this cell was in a late-stage of exflagellation, with 6 microgametes having already been released (Fig. 3E). Despite the late stage in exflagellation, there was still a wide variation in axoneme length in these 5 microgametocytes (Fig. 3H; average axoneme length $13.4 \pm 3.2 \, \mu m$ for exflagellating cells) compared with microgamete axonemes, which were less variable in length (Fig. 3H; average axoneme length $15.6 \pm 1 \, \mu m$; F-test $p = 0.03611$). The microgamete axonemes average length was longer than the exflagellating microgametocytes, suggesting that axoneme extension is still undergoing in microgametocytes at the exflagellation stage.

During exflagellation a portion of the microgametocyte nucleus containing a haploid genome is taken to form the microgamete nucleus[6,8,10,18]. However, little is known about how this genome inheritance is organised at the ultrastructural level. This was investigated further in microgametocytes undergoing exflagellation and in ex-flagellated microgametes. Intriguingly, our SBF-SEM imaging revealed that the nucleus is coiled around the axoneme of ex-flagellating and ex-flagellated microgametes, which has not been previously reported (Fig. 3C–G, I).

For improved resolution of the elongated nuclear structures, we performed dual axis cellular electron tomography of portions of exflagellated microgametes in order to understand how this unique coiling is organised (Fig. 4A, Supplementary Movie 4). This clearly showed nuclear structures tightly coiled around the outer doublet microtubules of the axoneme, all of which was enclosed within the flagellar membrane. There were distinctive focal electron densities within the membrane of the nucleus that were asymmetrically located close to the point where the nuclear envelope was closely associated with the outer doublet microtubules of the axoneme. This could represent the condensed chromatin indirectly crosslinked to axonemal microtubules through the nuclear envelope (Fig. 4A; asterisks, Supplementary Movie 4). The central point of each density was selected in each slice of the tomogram, which showed these densities to coil around the axoneme (Fig. 4B, C).

An important question is when coiling of the nucleus around the axoneme occurs. As shown in Fig. 3C–G, SBF-SEM reconstructions of microgametocytes at later stages of exflagellation there were many examples of flagella that were partially ex-flagellated. In the portion of flagellum still contained within the microgametocyte, we frequently observed the nucleus folding around the flagellum (Fig. 4D; inset E, F;

slice of SBF-SEM microgametocyte of 4D). Extensive analysis using thin-section transmission electron microscopy (Supple. Fig. 2) combined with cellular electron tomography revealed the nucleus interacting with the axoneme in a similar coiled arrangement (Fig. 4G, H) as seen in the microgamete (Fig. 4A). In order to investigate this further, we reconstructed a portion of the microgametocyte nucleus from a (serial) cellular electron tomogram showing the coil around the axoneme (Fig. 5; Supplementary Movie 5). Selected views of the model clearly shows a portion of the nuclear envelope to be extending out and coiling around the axoneme (Fig. 5A–C; inset C). Selected z-slices taken from this tomogram illustrates the nucleoplasm (Fig. 5D–I). This confirms that the lumen of the coils is continuous with that of the nucleoplasm. This data demonstrates that coiling of the nuclear material starts when axonemes are still within the microgametocyte.

Microgametes are motile and, like all flagellated cells, have distinct wave forms patterns. Coiling of the nuclear material around the axoneme might be important in maintaining this wave form. Therefore, having the nucleus elongated along the axoneme would be advantageous for motility. Taken together, these results suggest there is a highly organised periodic coiling of nuclear material around the axonemes within the microgamete and represents an extra-axonemal structure within microgametes of *Plasmodium berghei* which lack IFT[20].

The technology and development of SBF-SEM and cellular electron tomography has enhanced our understanding of how BB-NPs are organised in *Plasmodium* during male gamete formation and revealed an axonemal band around the nucleus, with axonemes coiled in different directions, showing chirality within this tight band. How axoneme assembly is restricted to this central band is not understood but might be influenced by the 3D organisation between BB and NPs in directing this process, along with inherent constraints in axoneme extension. In addition, tight positioning of axonemes in a central axonemal band could be due to localised accumulation of axoneme assembly proteins within this band. Furthermore, the organisation of axonemes in opposing orientations could generate momentum and rotational power to drive exflagellation.

The intricate coiling of the nucleus around the axoneme described here shows a level of ultrastructural complexity within the microgamete that has not previously been reported, and further work will be required to dissect the precise mechanism that drives coiling of the nucleus around the axoneme.

There are examples of other extra-axonemal structures associated with the microtubule axoneme in other eukaryotic organisms, such as the paraflagellar rod of some Kinetoplastida organisms[21] and the mitochondria of mammalian sperm, which coils around a portion of the axoneme[22]. The rapid process of microgamete generation is widely reported to be error-prone, and the detailed analysis shown here provides insights into the causes and complexity in the association between BBs/axonemes and nuclear structures in *Plasmodium* microgametocytes.

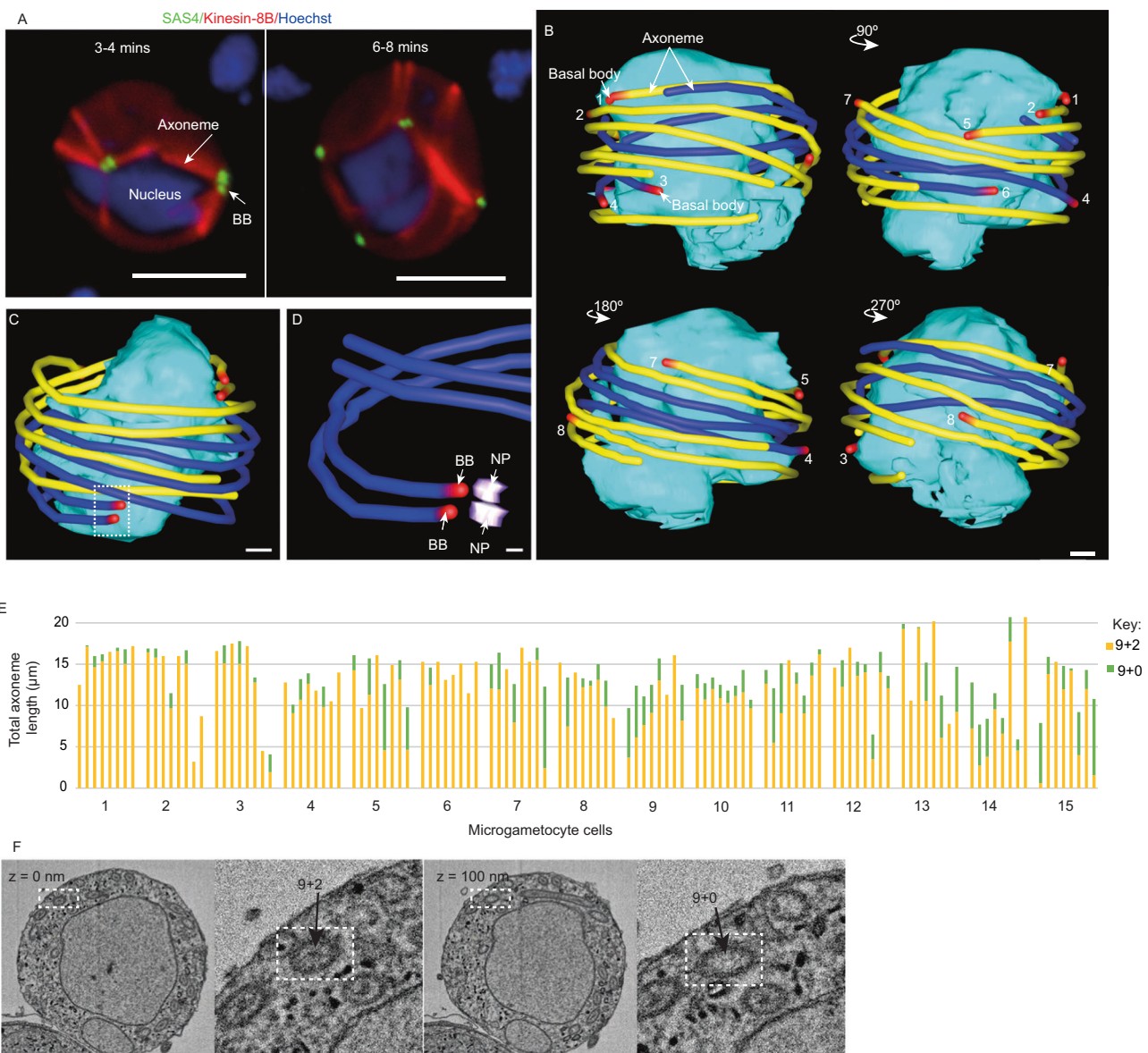

**Fig. 2 | Axonemes coil around the nucleus of the microgametocyte in opposite directions creating an axonemal band. A** Location of the basal body (BB) marker SAS4-GFP (green) in relation to axoneme marker kinesin-8B-mCherry (red) in male gametocytes at 3–4 min and 6–8 min post-activation, highlighting the formation of axonemes around the nucleus (Hoechst staining - blue). Scale bar = 5 μm. This experiment was repeated three times with similar results. **B** Microgametocyte 3D models (viewed at 90 °C rotations) from SBF-SEM data showing the axonemes forming a band around the nucleus. Axonemes are shown in yellow or blue depending on their direction of coiling around the nucleus (cyan), starting at the basal body (red). Axonemes are numbered to facilitate their identification in different model views. Scale bar = 500 nm. **C** Microgametocyte 3D model from SBF-SEM data showing the nucleus (cyan) surrounded by axonemes (blue/yellow) with paired basal bodies (red). Scale bar = 500 nm. **D** Inset showing the detail of Fig. 2C 3D model showing a basal body (BB - red) and nuclear pole (NP - purple) pair with their associated axonemes (blue) coiling in the same direction. Scale bar = 200 nm. **E** Bar graph showing the total length of individual axonemes in all pre-exflagellation microgametocytes found in our dataset ($n = 120$ axonemes in 15 cells). Axoneme lengths are divided into two sections – where the central pair is present (9 + 2, yellow) and where it is absent (9 + 0, green). Source data are provided as a Source Data file. **F** SBF-SEM data slices highlighting the presence of the outer doublets and the central pair (9 + 2, at $z = 0$ nm) and the presence of the outer doublets only (9 + 0) in a neighbouring section of the same axoneme ($z = 100$ nm). Scale bar = 500 nm. $n = 120$ axonemes.

## Methods

### Ethics statement

The animal work done at the University of Nottingham passed an ethical review process and was approved by the United Kingdom Home Office. Work was carried out under UK Home Office Project Licences (30/3248 and PDD2D5182) in accordance with the UK Animals (Scientific Procedures) Act 1986. Six- to eight-week-old female CD1 outbred mice from Charles River laboratories were used for all experiments. The conditions of mice kept were a 12 h light and 12 h dark (7 am till 7 pm) light cycle, the room temperature was kept between 20 and 24 °C and the humidity was kept between 40 and 60%.

### Generation of dual tagged parasite lines and live cell imaging

The C-terminus of SAS4 was tagged with GFP by single crossover homologous recombination in the parasite. To generate the SAS4-GFP line, a region of the *sas4* gene downstream of the ATG start codon was amplified using primers T2011 and T2012, ligated to p277 vector, and transfected by electroporation. The C terminus of kinesin-8B was

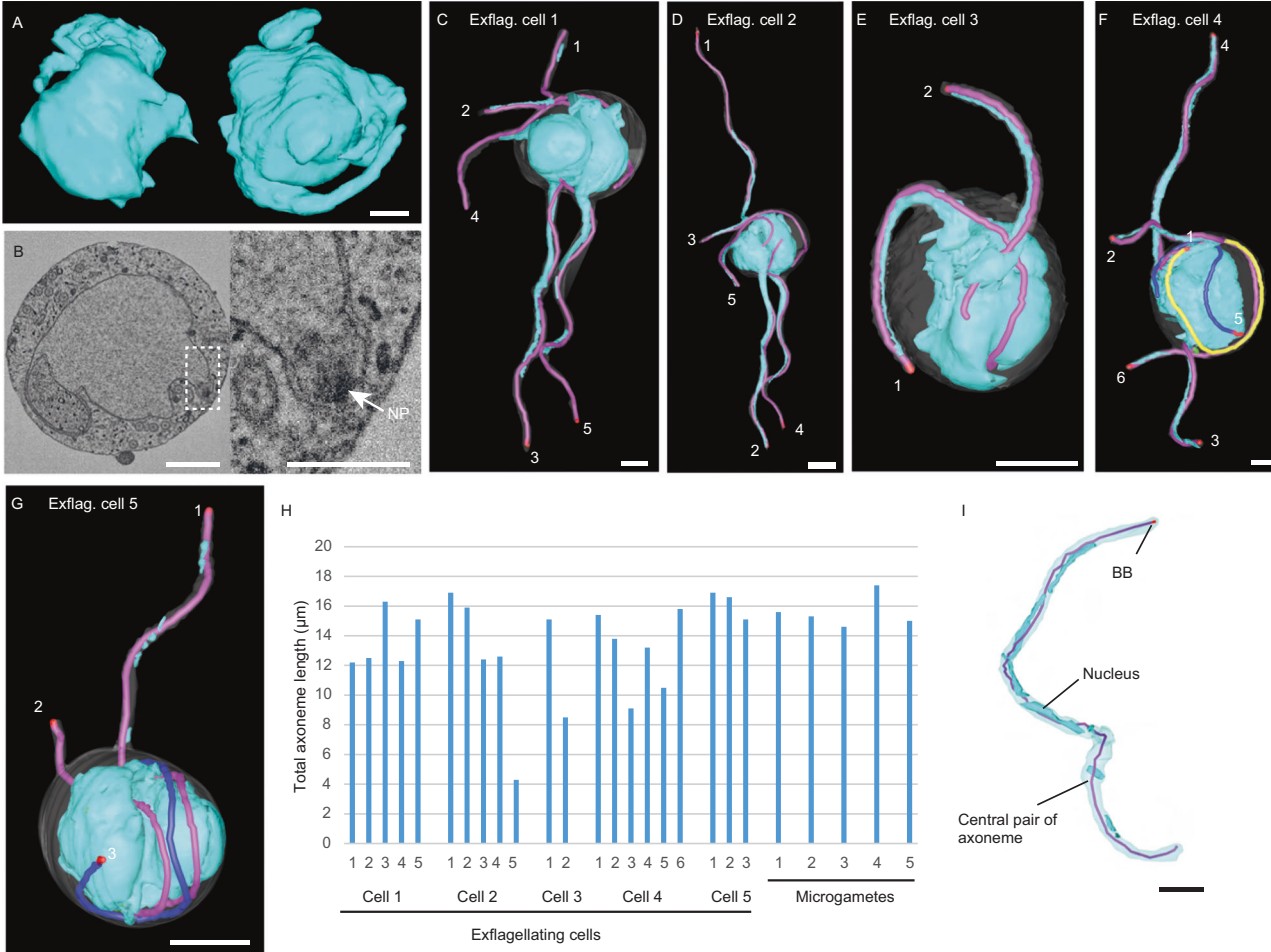

**Fig. 3 | A portion of the nucleus associated with each axoneme in exflagellating microgametocytes. A** Two different 3D models of nuclei (cyan) showing the complex 3D structure and nuclear protrusions. Scale bar = 1 μm. **B** SBF-SEM data slice of a nucleus showing the position of a nuclear pole (NP, dashed white box) within a nuclear projection. Scale bar = 1 μm. *n* = 15 microgametocyte cells. **C**–**G** 3D models of microgametocytes undergoing exflagellating at different stages. Nucleus (cyan), plasma membrane (white), basal body (red), exflagellating axoneme (pink), internally coiled axoneme (yellow/blue, depending on coiling direction). Axonemes are numbered to facilitate their identification in Fig. 3H. Scale bar = 1 μm. **H** Bar chart showing the axoneme lengths in exflagellating cells and microgametes. Exflagellating cell axonemes are numbered to facilitate their identification in Fig. 3C–G. Statistical F-test *p* = 0.03611. Source data are provided as a Source Data file. **I** 3D model of a free microgamete highlighting the ultrastructural features (basal body – red, central pair of the axoneme – pink, plasma membrane – pale blue, nucleus – cyan) with the nucleus coiling around the axoneme. Scale bar = 1 μm.

tagged with mCherry by single crossover homologous recombination in the parasite. To generate the kinesin-8B–mCherry line, a region of the *kinesin-8B* gene downstream of the ATG start codon was amplified using primers T1991 and T1992, ligated to p277 vector, and transfected. For mCherry tagging, a 1153 bp region of *Ndc80* without the stop codon was inserted upstream of the *mCherry* sequence in the p277 plasmid vector using KpnI and ApaI restriction sites. The p277 vector contains the human *dhfr* cassette, conveying resistance to pyrimethamine. Before transfection, the sequence was linearised using EcoRV. The *P. berghei* ANKA line 2.34 was used for transfection by electroporation. The oligonucleotides used to generate the tagged parasite lines are listed in Supplementary Table 1.

Parasite cells from lines expressing tagged SAS4-GFP (green) were mixed with cells from mCherry (red) tagged lines of the kinetochore marker NDC80[15] and the axoneme marker kinesin-8B[18] in equal numbers and injected into mice. Mosquitoes were fed on these mice 4 to 5 days after infection, when gametocytemia was high, and were checked for oocyst development and sporozoite formation at days 14 and 21 after feeding. Infected mosquitoes were then allowed to feed on naïve mice and after 4 to 5 days the mice were examined for blood stage parasitaemia by light microscopy of Giemsa-stained blood smears. Fluorescence microscopy showed that some parasites expressed both SAS4-GFP and NDC80-mCherry; or SAS4-GFP and kinesin-8B-mCherry in the resultant gametocytes. These gametocytes were purified as described above, activated, stained with Hoechst 33342, and then imaged (at desired time-points post-activation) by fluorescence microscopy using a 63x oil immersion objective on a Zeiss Axio Imager M2 microscope, fitted with an AxioCam ICc1 digital camera.

## Parasite culture and gametocyte purification

*Plasmodium berghei* transgenic lines expressing SAS4-GFP[15], and either kinesin-8B-mCherry[18] or NDC80-mCherry[23] were injected into phenylhydrazine treated mice[23]. Gametocyte enrichment was achieved by sulfadiazine treatment after 2 days of infection. The blood was collected on day 4 after infection and gametocyte-infected cells were purified on a 48% v/v NycoDenz (in PBS) gradient (NycoDenz stock solution: 27.6% w/v NycoDenz in 5 mM Tris-HCl, pH 7.20, 3 mM KCl, 0.3 mM EDTA). The gametocytes were harvested from the interface and activated in ookinete medium containing xanthurenic acid.

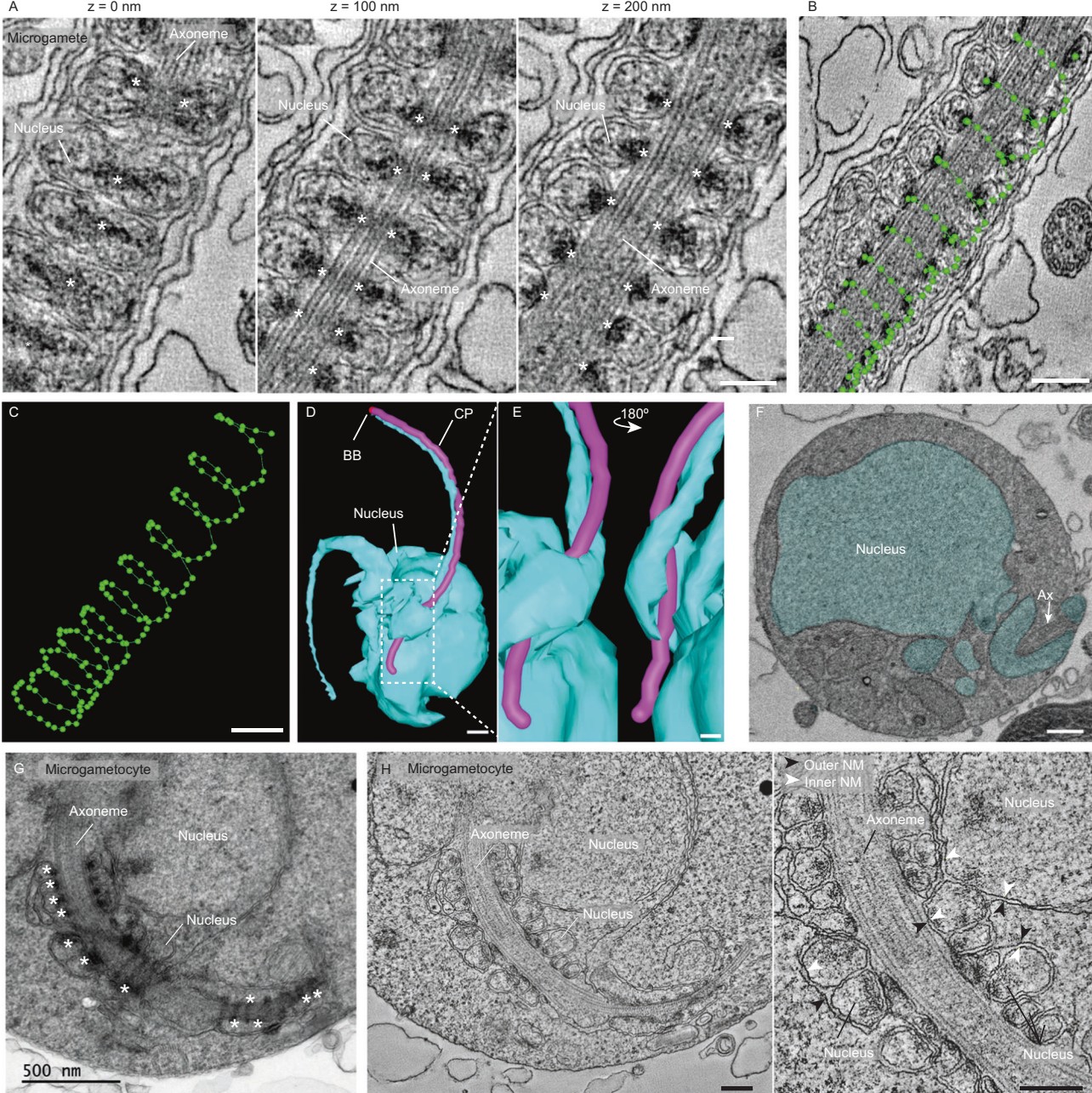

**Fig. 4 | Unique organisation of the nucleus coiling around the axoneme of microgametocytes and microgametes. A** Slices of a serial tomogram of a free microgamete showing the nucleus and electron dense material (asterisks) coiling around the axoneme. Scale bar = 200 nm. *n* = 6 microgametes. **B, C** 3D reconstruction of the coiled path (green) described by the electron dense structures inside the coiled nucleus of the free microgamete shown in Fig. 4A. Scale bar = 200 nm. 3D model (**D, E**) and SBF-SEM image (**F**) of an exflagellating microgametocyte showing a section of the nucleus (blue) coiling around the central pair (pink) of the axoneme (Ax). Basal body (BB - red) Scale bar = 500 nm (**D**) and 200 nm (**E** and **F**). The images in (**E**) show two higher magnification views of the model in the area dashed in (**D**), at 180 °C rotations. Axoneme (Ax). **G** Transmission electron micrograph of a semi-thin (150 nm) section highlighting the coiling of the nucleus around the axoneme within a microgametocyte. Asterisks (white) highlight the electron dense areas that were also seen in the coiled nucleus of free microgametes. Scale bar = 500 nm. **H** Slice of a tomogram of the area in Fig. 4G, showing that the coils around the axoneme are delimited by nuclear membranes. White arrows indicate the inner nuclear membrane (INM) and black arrows indicate the outer nuclear membrane (ONM). Scale bar = 200 nm. *n* = 15 microgametocyte cells.

## Serial block face scanning electron microscopy (SBF-SEM) of culture derived gametocytes

Fractions enriched in *Plasmodium berghei* gametocytes (produced as described above) were fixed at room temperature in 2.5% glutaraldehyde in 0.1 M phosphate buffer, 15 min after activation, to allow for an asynchronous population containing the different stages of gametogenesis and exflagellation to be analysed. Samples were then spun and washed three times in 0.1 M phosphate buffer, and post-fixed in 1% osmium tetroxide in 1.5% potassium ferrocyanide in 0.1 M phosphate buffer (for 45 min at room temperature, and in the dark). After the first osmium step, samples were washed three times in 0.1 M phosphate buffer, incubated in 1% tannic acid in 0.1 M phosphate buffer for 30 min at room temperature, and then subjected to a second osmium step (2% $OsO_4$ in $ddH_2O$, for 30 min, at room temperature, and in the dark). Samples were then incubated in 2% uranyl acetate in $ddH_2O$ for 2 h, dehydrated in acetone (a progressive series of acetone

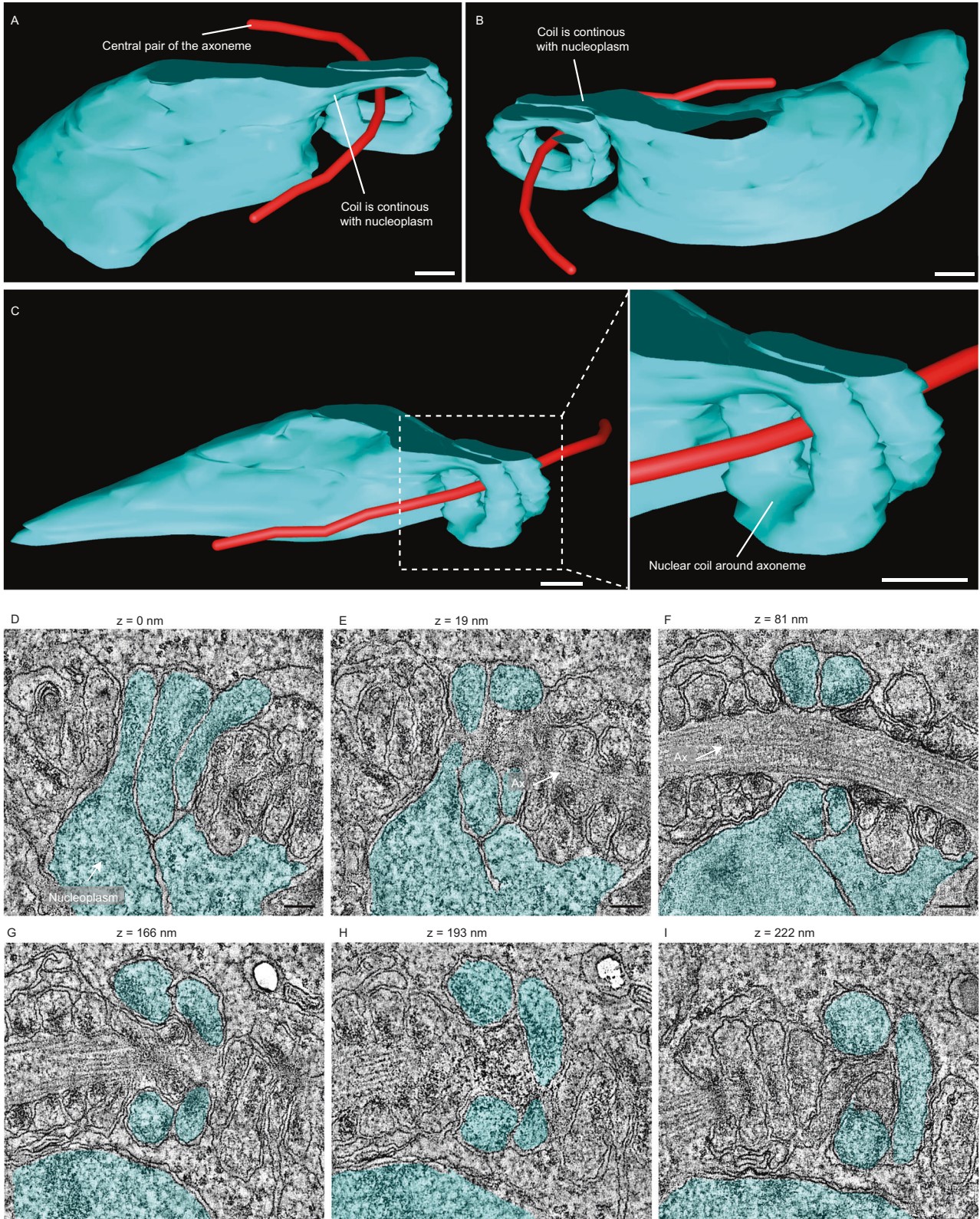

**Fig. 5 | Coiling of the nucleus around the axoneme is observed inside the microgametocyte. A–C** 3D model showing a portion of the nucleoplasm (cyan) coiling around the axoneme (represented by the path of the central pair of microtubules, in red). Scale bar = 100 nm. **D–I** Slices of a serial tomogram of a microgametocyte, showing a nuclear projection surrounding the axoneme (Ax), with the nucleoplasm (cyan) forming a coil around the axoneme. Scale bar = 100 nm. $n = 3$ microgametocytes cells. Each slice represents a z slice of 1 nm.

concentrations from 20%, 40%, 90% to 100% with three changes in molecular-sieved ultradry acetone over 4 h) and embedded in TAAB 812 Hard resin (TAAB, catalogue number T030). The tips of resin blocks containing samples were trimmed and mounted onto aluminium pins using conductive epoxy glue and silver dag, and then sputter coated with a layer (10–13 nm) of gold, in an Agar Auto Sputter Coater (Agar Scientific). Before SBF-SEM imaging, ultrathin sections (70 nm) of the block face were examined in a Jeol JEM 1400 Flash transmission electron microscope (JEOL), to verify sample quality. Samples were then imaged in a Merlin VP compact high resolution scanning electron microscope (Zeiss) equipped with a 3View stage (Gatan-Ametek), and an OnPoint back-scattered electron detector (Gatan-Ametek), in variable pressure. The following imaging conditions were used: 3 kV, 30 μm aperture, 30 pascal variable pressure, 3 nm pixel size, 4 μs pixel time, 100 nm section thickness.

### SBF-SEM and tomography data segmentation and analysis
Data were processed using the IMOD software package[11]. Briefly, image stacks were assembled, corrected (for z scaling and orientation) and aligned using eTOMO, and 3D models were produced using 3dmod. Whole individual microgametocyte cells with internally coiled axonemes ($n = 15$ cells), microgametocytes undergoing exflagellation ($n = 5$ cells), and free microgametes ($n = 5$ cells) were identified and manually segmented from trimmed regions of original datasets. The microgametocytes and microgametes analysed here represented all entire cells of each type found in the SBF-SEM datasets (incomplete cells were discarded from the analysis). In each microgametocyte and microgamete, the cell membrane, the nucleus, the axonemes with subtending basal bodies and the nuclear poles (microgametocyte only) were identified based on distinctive ultrastructural features, and then segmented manually as individual objects (Supplementary Fig. 1). For the cell body and nucleus, the outer membrane was used to define the edges of segmentation. Axonemes were identified by the outer microtubule doublets as an electron dense circle with the central pair appearing as a central electron density within this electron dense circle (9 + 2) or without the central pair present (9 + 0). Basal bodies were segmented as an electron dense structure at the proximal end of the axoneme. Nuclear poles were segmented as the area that included all electron dense structures known to form the NP. Volume, surface area and length measurements for different structures were obtained in 3dmod, based on objects surface rendering. Statistical analysis of axoneme length variability (F-test) was calculated in Microsoft Excel.

### Dual axis cellular electron tomography
Fractions enriched in *Plasmodium berghei* gametocytes (produced as described above) were fixed in 4% glutaraldehyde in 0.1 M phosphate buffer and processed for electron microscopy as described previously[24]. Briefly, samples were post-fixed in osmium tetroxide, treated en bloc with uranyl acetate, dehydrated in acetone and embedded in Spurr's epoxy resin. Thin sections were stained with uranyl acetate and Reynolds' lead citrate prior to examination in a JEOL JEM-1400 Flash transmission electron microscope (JOEL, UK). Serial-section cellular electron tomography (ssET) was performed on 150 nm sections collected onto formvar-coated slot grids. Grids were mounted in a Fishione dual-axis tomography holder (Fischione instruments) and dual-axis tilt-series (55º to −55º, with 1º tilt between images) of the same area in consecutive sections were acquired using SerialEM[25]. Tomogram generation and serial tomogram joining were perfomed in ETomo[11].

### Statistics and reproducibility
No statistical method was used to predetermine sample size. Statistical analysis of axoneme length variability (F-test) was calculated in Microsoft Excel. The microgametocytes and microgametes analysed

here represented all entire cells of each type found in the SBF-SEM datasets. Whole individual cells were analysed. Incomplete cells were discarded from the analysis. The experiments were not randomised. The Investigators were not blinded to allocation during experiments and outcome assessment.

### Reporting summary
Further information on research design is available in the Nature Portfolio Reporting Summary linked to this article.

## Data availability
The SBF-SEM datasets generated and analysed for this study have been deposited and are available in the EMPIAR repository database[26] under the EMPIAR ID: EMPIAR -11710. The supplementary movies generated for this study are available at Zenodo accession code 8416867. Links to Source data are provided with this paper.

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

## Acknowledgements

We would like to thank The Oxford Brookes Centre for Bioimaging for assistance in carrying out this project. M.H. was funded by a Nigel Groome studentship. We thank Declan Brady at University of Nottingham for technical assistance in mouse and parasite work. This work was supported by MRC UK (MR/K011782/1) to R.T. and BBSRC (BB/N017609/1) and ERC advance grant funded by UKRI Frontier Science (EP/X024776/1) to R.T. and M.Z.

## Author contributions

M.H. coordinated the project, performed data generation of volume-EM, 3D modelling of volume-EM collection, analysis and helped in writing the manuscript; F.M.L. performed data generation of volume-EM and provided significant manuscript editing; D.J.P.F. coordinated the project, performed data collection of volume-EM, data analysis and provided significant manuscript editing; M.Z. perform data generation of live cell imaging and mice-related work as well as final manuscript editing; S.V. and R.T. conceived, coordinated the project and prepared the manuscript. All authors contributed to manuscript editing and revisions.

## Competing interests

The authors declare no competing interests.
