## [Peer Review File · Nature Communications]

Atypical flagella assembly and haploid genome coiling during male gamete formation in *Plasmodium*REVIEWER COMMENTS

Reviewer #1 (Remarks to the Author):

The work by Molly Hair and colleagues presented in this manuscript provides an unprecedented, detailed account of the morphological peculiarities of *Plasmodium berghei*'s male gametocytes. This work advances the ultrastructural understanding of how flagella are formed, coiled, and connected to the nucleus leading up to ex-flagellation. The manuscript provides an interesting discussion and poses hypotheses on the functional advantages on the peculiar topological arrangements described.

I have no major suggestions for improvement. However, I have a few minor suggestions to improve the manuscript's clarity.

I would suggest slightly rewording the abstract.

For example, it states that sexual reproduction is essential for gametogenesis. I believe the authors mean to state that the events occurring within the *Anopheles* mosquito are essential for gametogenesis, and that this is one vital part of sexual reproduction. However, as written, the sentence is confusing. The sentence "To better understand the 3D organization male gametogenesis we used serial block face scanning... "seems to be missing some words ("leading to" perhaps?).

Line 116: I would suggest rewording "clearly associated" for "physically adjacent" Topological data does not determine association.

Line 143: substitute "between" for "among"

Line 152 reads "microgametes are motile flagella" please reword for clarity

Line 280: please correct the term "Kinetoplastia" (Kinetoplastea)

Reviewer #3 (Remarks to the Author):

The manuscript submitted by Molly Hair and colleagues utilizes serial block-face scanning electron microscopy and cellular electron tomography to examine the three-dimensional ultrastructure of *P. berghei* microgametocytes and male gamete formation. This manuscript provides novel data on axoneme formation, orientation, and interactions with the nucleus in addition to providing novel insights into the process of exflagellation. Overall, this manuscript improves our understanding of the ultrastructural complexities of these processes. While the findings presented by the authors are interesting and well presented, there are a number of concerns that this reviewer believes should be addressed. The primary concerns consist of insufficient descriptions of the cell selection and reconstruction process, and an inability to properly assess the image quality and resolution of the SBF-SEM image series to ensure that they are sufficient to inform the 3D reconstructions presented.

1. SBF-SEM segmentation – The authors state that the cell membrane, nucleus, axonemes and nuclear poles were identified using distinctive ultrastructural features; however, only the features used for nucleus and nuclear pole segmentation were specified. Similarly, the ultrastructural features used for microgamete reconstruction should be clearly defined in the text.

2. The authors state that from the datasets, 15 individual whole microgametes with their axonemes internally coiled were identified and analyzed. Do these 15 microgametes represent the totality of the microgametes in the dataset, or do they represent a subpopulation of imaged microgametes? If the latter, it would be necessary to describe the methods by which the cells were randomly and systematically selected for analysis.

3. Lines 120-124 – How was the mid-point of each electron dense structure determined? The average distance between associated BBs and NPs was stated, but the standard deviation was not provided.

4. Movie 1: Rendering/compression artifacts make it difficult to assess image quality and resolution. As is, it is not clear that the dataset contains the resolution necessary for

reconstruction of the structures described.

5. Movie 2: It would be necessary to play through the image series without the reconstruction present to allow readers to assess cellular ultrastructure and verify accuracy of the 3D reconstruction. In addition, the scale bar appears to reset at the 9 second mark.

6. Movie 3: The same can be said for Movie 3; playing through the image series in the absence of the 3D reconstruction allows readers to assess cellular ultrastructure and verify accuracy of reconstruction. The scale bar appears to reset at the 14 second mark.

7. Additional details on SBF-SEM tissue preparation would be desired. For example: duration and temperature of fixation in 2.5% glutaraldehyde and an accounting of the acetone dehydration steps.

8. Figure 1, Panel A: While the authors state that an arrow identifies "filamentous material," two arrows are used in the first portion of this figure which identify two different structures. While the authors' intentions are relatively clear, this should be rectified for clarity (Use of an arrowhead, for example).

9. Figure 3, Panel H: This graph shows axoneme lengths in exflagellating cells and microgametes. Consider providing average axoneme length for microgametes (\pm standard deviation) in addition to the exflagellating cell average provided in the text.

10. Figure 4, Panels G & H: In this reviewer's opinion, the asterisks in these panels obscure interesting details in each image and distract from the stated purpose of the panels. Suggest removal.

11. An example micrograph of a complete microgamete taken from the SBF-SEM or transmission electron tomography images would be a welcome addition to the figures.

Reviewer #4 (Remarks to the Author):

The authors of this manuscript used fluorescence microscopy, serial block face imaging and electron tomography to investigate the nuclear and axonemal architecture during male *P. Bergheii* gametogenesis. This is important work showing detailed ultrastructural analysis of this complex process. However, there is insufficient evidence to claim that the helical membrane compartment in Fig. 4 is the nucleus. The electron dense structures contained within are indeed consistent with condensed genome staining, but no evidence is shown that the lumen is continuous with the nucleoplasm. Supplemental movie 5 shows that these membrane bound structures are contained within what is likely an extension of the nuclear envelope. Topologically, this would place them in the perinuclear space. If this really the case, it would make the observation even more fascinating. The serial block face data does not support the presented hypothesis either: nuclear protrusions are generally wider and do not wind around axonemes as tightly.

Hopefully, this can be clarified by revisiting existing data, considering that new specimens would require mice to be sacrificed. Alternatively, the conclusion could be changed to reflect the data.

Minor comments:

Line 27 and 30

Consider using "elaborate" instead of "exquisite"

Line 218: "axonemes seemed less variable in length"

There are statistical tools to determine whether they are or they are not.

Line 205 "There were multiple projections"

Consider using "protrusions" instead of "projections" as the latter is often used to mean image projection

Line 207 "but this was not significant"

The method used to determine significance should be stated.

REVIEWER COMMENTS

Reviewer #1 (Remarks to the Author):

The work by Molly Hair and colleagues presented in this manuscript provides an unprecedented, detailed account of the morphological peculiarities of *Plasmodium berghei*'s male gametocytes. This work advances the ultrastructural understanding of how flagella are formed, coiled, and connected to the nucleus leading up to ex-flagellation. The manuscript provides an interesting discussion and poses hypotheses on the functional advantages on the peculiar topological arrangements described.

I have no major suggestions for improvement. However, I have a few minor suggestions to improve the manuscript's clarity.

I would suggest slightly rewording the abstract.

For example, it states that sexual reproduction is essential for gametogenesis. I believe the authors mean to state that the events occurring within the *Anopheles* mosquito are essential for gametogenesis, and that this is one vital part of sexual reproduction. However, as written, the sentence is confusing. The sentence "To better understand the 3D organization male gametogenesis we used serial block face scanning... "seems to be missing some words ("leading to" perhaps?).

Thank you for your suggestions, comments and enthusiasm for our work. We have modified the abstract accordingly in Lines 21-26.

Line 116: I would suggest rewording "clearly associated" for "physically adjacent" Topological data does not determine association.

We have reworded "clearly associated" for "physically adjacent" in now Line 117.

Line 143: substitute "between" for "among".

We have reworded "between" for "among" now in Line 146.

Line 152 reads "microgametes are motile flagella" please reword for clarity.

We have reworded "microgametes are motile flagella" to "Microgametes are motile, and, like all flagellated cells" in now Line 265 for clarity.

Line 280: please correct the term "Kinetoplastia" (Kinetoplastea).

We have corrected the term 'Kinetoplastid' to 'Kinetoplastida' in now Line 293.

Reviewer #3 (Remarks to the Author):

The manuscript submitted by Molly Hair and colleagues utilizes serial block-face scanning electron microscopy and cellular electron tomography to examine the three-dimensional ultrastructure of *P. berghei* microgametocytes and male gamete formation. This manuscript provides novel data on axoneme formation, orientation, and interactions with the nucleus in addition to providing novel insights into the process of exflagellation. Overall, this manuscript improves our understanding of the ultrastructural complexities of these processes. While the findings presented by the authors are interesting and well presented, there are a number of concerns that this reviewer believes should be addressed. The primary concerns consist of insufficient descriptions of the cell selection and reconstruction process, and an inability to properly assess the image quality and

resolution of the SBF-SEM image series to ensure that they are sufficient to inform the 3D reconstructions presented.

1. SBF-SEM segmentation – The authors state that the cell membrane, nucleus, axonemes and nuclear poles were identified using distinctive ultrastructural features; however, only the features used for nucleus and nuclear pole segmentation were specified. Similarly, the ultrastructural features used for microgamete reconstruction should be clearly defined in the text.

Thank you for highlighting this. We have defined in our methodology section Line 288-393 how we have identified the cell membrane, axonemes and microgametocyte ultrastructure's. We have also included as a supplementary figure (Supp. Fig. 1) images of the diagnostic ultrastructural features used to identify the plasma membrane, nucleus, nuclear poles, axonemes (both 9+2 and 9+0) and basal bodies, for the manual segmentation of microgametocytes and microgametes. For additional clarity, we have also improved the resolution of the movies, and labelled features in the movies to facilitate identification, and highlight the quality of the data. We have also submitted the original SBF-SEM datasets to the Electron Microscopy Public Image Archive (EMPIAR) and the movies to Zenodo, so that the data quality can be easily verified by readers. The SBF-SEM datasets generated and analysed for this study are available in the EMPIAR repository, EMPIAR ID: EMPIAR -11710. The supplementary movies generated for this study are available at Zenodo <https://zenodo.org/record/8416867>.

2. The authors state that from the datasets, 15 individual whole microgametes with their axonemes internally coiled were identified and analyzed. Do these 15 microgametes represent the totality of the microgametes in the dataset, or do they represent a subpopulation of imaged microgametes? If the latter, it would be necessary to describe the methods by which the cells were randomly and systematically selected for analysis.

Thank you for your comment. The cells described in this manuscript represent all entire microgametes and microgametocytes found in the datasets. The only selection criterion was that the cells had to be completely contained within the imaged volume. As these cells are large (particularly the exflagellating microgametocytes), the datasets contained parts of many other microgametes and microgametocytes; however, these were incomplete and could not be used in the analysis. We have included this in our results section to clarify this in Lines 94-95 and methodology section 'SBF-SEM segmentation and data analysis' Lines 382-384.

3. Lines 120-124 – How was the mid-point of each electron dense structure determined? The average distance between associated BBs and NPs was stated, but the standard deviation was not provided.

Thank you for your comment. Basal bodies and nuclear poles typically spanned across 2-3 (for basal bodies) and 3-4 (for nuclear poles) z slices of the SBF-SEM

datasets. The mid-point of each electron dense structure (basal body and nuclear pole) was considered as the center of the electron density of either the basal body or nuclear pole. We have included this description within our results section Line 124-126. We have also added the standard deviation to the distance between the associated BB and NPs in Line 127. We have also added a black dashed line into Figure 1D to show the average distance between associated NP and BB's.

4. Movie 1: Rendering/compression artifacts make it difficult to assess image quality and resolution. As is, it is not clear that the dataset contains the resolution necessary for reconstruction of the structures described.

Thank you for highlighting the image quality of the movies. We have now improved the movies to highlight the quality of the dataset. uploaded our movies to a repository recommended by Nature Communications so you can see the full movie with high resolution Given the size constraints of the movies, we provide in the revised version of the manuscript a lower resolution version of both datasets in movie 1, but we also uploaded our SBF-SEM datasets in full resolution into the Electron Microscopy Public Image Archive (EMPIAR), and movies to Zenodo, to facilitate image quality assessment.

5. Movie 2: It would be necessary to play through the image series without the reconstruction present to allow readers to assess cellular ultrastructure and verify accuracy of the 3D reconstruction. In addition, the scale bar appears to reset at the 9 second mark.

Thank you for your comment on Movie 2. We have re-created Movie 2 to also include both the original SBF-SEM dataset and segmentation of the microgametocyte cell. We have also added labels and highlighted key ultrastructural features in the movie so readers can assess the cellular ultrastructure and see how the 3D model is built on top of the image series. Thank you for bringing to our attention that the scale bar resets at the 9 second mark. We have amended this.

6. Movie 3: The same can be said for Movie 3; playing through the image series in the absence of the 3D reconstruction allows readers to assess cellular ultrastructure and verify accuracy of reconstruction. The scale bar appears to reset at the 14 second mark.

Thank you for your comment on Movie 3. We have re-created Movie 3 to also include both the original SBF-SEM dataset and segmentation of the exflagellating microgametocyte cell. We have also added labels and highlighted key ultrastructural features in the movie so readers can assess the cellular ultrastructure and see how the 3D model is built on top of the image series Thank you for bringing to our attention that the scale bar resets at the 14 second mark. We have amended this.

7. Additional details on SBF-SEM tissue preparation would be desired. For

example: duration and temperature of fixation in 2.5% glutaraldehyde and an accounting of the acetone dehydration steps.

Thank you for your suggestion. We have added this to our methodology in Lines 349-350 and 360-361.

8. Figure 1, Panel A: While the authors state that an arrow identifies “filamentous material,” two arrows are used in the first portion of this figure which identify two different structures. While the authors’ intentions are relatively clear, this should be rectified for clarity (Use of an arrowhead, for example).

Thank you for your comment. We have now used a single white arrowhead to point to the filamentous material between the nuclear poles and basal body and updated the figure legend to reflect this.

9. Figure 3, Panel H: This graph shows axoneme lengths in exflagellating cells and microgametes. Consider providing average axoneme length for microgametes (\pm standard deviation) in addition to the exflagellating cell average provided in the text.

Thank you for your suggestion. We have added the average axoneme length and standard deviation for the ex-flagellating cells and microgametes in Lines 219 and 221.

10. Figure 4, Panels G & H: In this reviewer’s opinion, the asterisks in these panels obscure interesting details in each image and distract from the stated purpose of the panels. Suggest removal.

Thank you for your suggestion. We have removed some and re-positioned these asterisks on Figure 4G and removed them entirely on Figure 4H.

11. An example micrograph of a complete microgamete taken from the SBF-SEM or transmission electron tomography images would be a welcome addition to the figures.

Thank you for your suggestion. We have created a Supplementary Figure 2 which includes random thin sections of transmission electron micrographs of microgametes that illustrate the coiling of the nucleus the axoneme. In our Supplementary Figure 1, we have also included SBF-SEM slices and serial sections of a microgamete showing the microgamete ultrastructural features.. Unfortunately, due to the 3D nature of the raw data it is not possible to easily show a full-length microgamete in a single image, so we hope this is what the reviewer was hoping for. Fig 3I is a 3D model of a microgamete from our SBF-SEM dataset, which represents at least 20 z-slices. In addition, the reader can now visualize the SBF-SEM datasets and uncompressed movies to observe complete microgametes via EMPIAR (EMPIAR -11710).

Reviewer #4 (Remarks to the Author):

The authors of this manuscript used fluorescence microscopy, serial block face imaging and electron tomography to investigate the nuclear and axonemal architecture during male *P. bergheii* gametogenesis. This is important work showing detailed ultrastructural analysis of this complex process. However, there is insufficient evidence to claim that the helical membrane compartment in Fig. 4 is the nucleus. The electron dense structures contained within are indeed consistent with condensed genome staining, but no evidence is shown that the lumen is continuous with the nucleoplasm. Supplemental movie 5 shows that these membrane bound structures are contained within what is likely an extension of the nuclear envelope. Topologically, this would place them in the perinuclear space. If this really the case, it would make the observation even more fascinating. The serial block face data does not support the presented hypothesis either: nuclear protrusions are generally wider and do not wind around axonemes as tightly.

Hopefully, this can be clarified by revisiting existing data, considering that new specimens would require mice to be sacrificed. Alternatively, the conclusion could be changed to reflect the data.

Thank you very much for your appreciation of our work. We appreciate your comment on whether we have sufficiently supported our conclusion ‘that the lumen is continuous with the nucleoplasm’. We understand your concerns, so we analysed an additional serial tomogram of an axoneme within a microgametocyte prior to exflagellation. By modelling part of this serial tomogram, it is clear that a nucleoplasm protrusion is continuous with the coils observed around the axoneme (Figure 5 and movie 5).

The identification of specific components of the coil should, in future studies, help clarify when and how the coil is formed, and where it is formed along the length of the axoneme. Nevertheless, we believe that in this study we can still include that our datasets provide good evidence that previously undescribed nuclear coils are found both in microgametocytes and microgametes of *Plasmodium berghei*, based on the clear identification of these structures in tomograms.

Minor comments:

Thank you for your suggestions.

Line 27 and 30. Consider using “elaborate” instead of “exquisite”

We have changed ‘exquisite’ to ‘elaborate’ in Line 27.

Line 218: “axonemes seemed less variable in length”

There are statistical tools to determine whether they are or they are not.

We have performed statistical testing on microgametocyte and microgamete axoneme length variability (results are mentioned in Line 222, and the statistical method is mentioned in Line 398 of our methodology section).

Line 205 “There were multiple projections”

Consider using “protrusions” instead of “projections” as the latter is often used to mean image projection

We have changed ‘projections’ to ‘protrusions’ in Line 207.

Line 207 “but this was not significant”. The method used to determine significance should be stated.

We have stated the method of statistical testing in Line 397-398.

We thank the reviewers for their comments and suggestions of our work. In addition, we have also made minor editorial changes throughout the manuscript.

REVIEWERS' COMMENTS

Reviewer #3 (Remarks to the Author):

This revised manuscript submitted by Molly Hair and colleagues fully addresses this reviewer's comments and concerns. In this reviewer's opinion, the revisions improve the transparency and reproducibility of the methods and results and enhance the clarity of the data and figures presented.

Reviewer #4 (Remarks to the Author):

The revised manuscript by Molly Hair and colleagues sufficiently addresses the previously ambiguous topology of axonemal coils in figure 5 and supplemental movie 5. I also appreciate the changes made to the other supplemental videos.

Minor comments:

30-31

"exquisitely complex structure"

I would suggest replacing or omitting the word "exquisite", as it is a subjective judgement of quality. I would also recommend having a brief look at "Use of positive and negative words in scientific PubMed abstracts between 1974 and 2014: retrospective analysis" (doi: <https://doi.org/10.1136/bmj.h6467>) which outlines why the use of positive words in scientific writing might be undesirable.

207-209

There were multiple protrusions with BB/NPs often positioned at the end of these projections (Fig 3B; arrow).

In my previous comment I suggested using "protrusions" instead of "projections" due to the latter being typically used to mean image projection (especially in electron microscopy). Both are likely correct, but in any case I would suggest being consistent in the use of either within this sentence and throughout the rest of the manuscript.

REVIEWER COMMENTS

Reviewer #3 (Remarks to the Author):

This revised manuscript submitted by Molly Hair and colleagues fully addresses this reviewer's comments and concerns. In this reviewer's opinion, the revisions improve the transparency and reproducibility of the methods and results and enhance the clarity of the data and figures presented.

Thank you for comments. We are pleased that the changes made to the manuscript regarding improving the resolution and clarity of the data satisfy your previous comments.

Reviewer #4 (Remarks to the Author):

The revised manuscript by Molly Hair and colleagues sufficiently addresses the previously ambiguous topology of axonemal coils in figure 5 and supplemental movie 5. I also appreciate the changes made to the other supplemental videos.

Thank you for your comments. We are pleased that the data shown in Figure 5 addresses your previous comments regarding the axonemal coils.

Minor comments:

30-31

"exquisitely complex structure"

I would suggest replacing or omitting the word "exquisite", as it is a subjective judgement of quality. I would also recommend having a brief look at "Use of positive and negative words in scientific PubMed abstracts between 1974 and 2014: retrospective analysis" (doi: <https://doi.org/10.1136/bmj.h6467>) which outlines why the use of positive words in scientific writing might be undesirable.

Thank you for your comment and suggestion. We have removed the word "exquisite".

207-209

There were multiple protrusions with BB/NPs often positioned at the end of these projections (Fig 3B; arrow).

In my previous comment I suggested using "protrusions" instead of "projections" due to the latter being typically used to mean image projection (especially in electron microscopy). Both are likely correct, but in any case I would suggest being consistent in the use of either within this sentence and throughout the rest of the manuscript.

Thank you for your comment. We have used "protrusions" instead of "projections" and have checked to ensure this is consistent throughout the rest of the manuscript.

We thank the reviewers for their comments and suggestions of our work.